# Photochemical C3-amination of pyridines via Zincke imine intermediates

Kitti Franciska Szabó[1,4], Piotr Banachowicz [1,4], Antoni Powała[1,2], Danijela Lunic[3], Ignacio Funes Ardoiz [3] ✉ & Dorota Gryko [1] ✉

Selective skeletal and peripheral editing of the pyridine moiety has broadly expanded the chemical space. While C-H functionalization at C2 and C4 positions are enabled by the inherent reactivity of this heteroarene, selective derivatization at the C3 position has long posed a significant challenge. Recently, based on a dearomatization-rearomatization sequence, involving Zincke imine intermediates, selective halogenation (-Br, -Cl, and -I) and isotopic labelling were accomplished. Here, we report a mild and regioselective method for C3-amination that relies on the photochemical reaction of Zincke imine with an amidyl radical generated from N-aminopyridinium salts. Mechanistic and theoretical studies indicate that radical intermediates are involved and explain the C3 regioselectivity of the reaction.

Heterocyclic scaffolds are present in numerous natural products, pharmaceuticals, agrochemicals, and have found their place in material science[1-3]. Among them, the pyridine moiety is, according to the US Food and Drug Administration, one of the most common motif in approved drugs (Fig. 1A)[4,5]. As biological activity can be fine-tuned by the substitution pattern at this core, methodologies for pyridine functionalizations are highly sought.

Numerous strategies for the peripheral editing of pyridines have been developed but, due to their inherent reactivity functionalization at C2 and C4 positions of the heterocycle prevail[6-9], with chemical modification at the C3 position still presenting a significant challenge. Classical C3-halogenation or C3-nitration via electrophilic aromatic substitutions suffer from harsh reaction conditions, excessive use of the desired pyridine, and low regioselectivity (Fig. 1B)[10,11]. Over the last decades, peripheral editing methodologies have emerged as a powerful tool for the introduction of substituents at the C3 position, primarily employing non-directed transition-metal catalysis[12-17], directing groups[7], and most recently a dearomatization-aromatization strategy (Fig. 1B)[9]. In this approach, electron-deficient azines are transformed into electron-rich intermediates (enamine, dienamine) that react with electrophilic reagents. Subsequent rearomatization leads to C3-substituted pyridine derivatives. Along this line, Wang and coworkers reported a one-pot borane-catalysed pyridine hydroboration tandem reactions[18]. Herein, a nucleophilic 1,4-dihydropyridine (1,4-

DHP) formed, reacts with electrophiles (imines, enol esters, SCF_3-, SCF_2H-, -CN reagents) and undergoes subsequent oxidation to afford C3-functionalized pyridines[19-23]. In 2022, Studer and coworkers developed an efficient C3-functionalizations of pyridines involving a bench-stable electron-rich oxazinopyridine generated from dimethyl acetylenedicarboxylate, methyl pyruvate and a pyridine[24,25]. This intermediate, as a dienamine, reacts with electrophilic reagents or radicals leading to, after rearomatization, C3-functionalized pyridines. By switching from an oxazino intermediate to a pyridinium salt, direct selective peripheral editing of pyridines at C4 was achieved[26-28]. The McNally group proposed another strategy for C3-halogenation of pyridines that proceeds via ring opening, halogenation, and ring-closure[29] for which they modified the classic Zincke reaction[30,31]. Furthermore, they expanded their methodology to synthesize N-(heteroaryl)pyridinium salts[32], and to incorporate stable radioisotope [15]N atom into the pyridine ring[33].

In recent years, significant advances have been made in photoredox catalysis and have expanded the toolbox of radical transformations available to synthetic chemists[34-36]. Recently, we and others have developed photochemical methods for site-selective amination of electron-rich double bonds using electrophilic N-centred radicals[37-40].

Based on reports concerning the functionalization of Zincke imines[41-47], and our studies on photochemical amination of dienol

[1]Institute of Organic Chemistry Polish Academy of Sciences, Warsaw, Poland. [2]Department of Chemistry, Warsaw University of Technology, Warsaw, Poland. [3]Departamento de Química, Instituto de Química de la Universidad de La Rioja, Universidad de La Rioja, Logroño, Spain. [4]These authors contributed equally: Kitti Franciska Szabó, Piotr Banachowicz. ✉e-mail: ignacio.funesa@unirioja.es; dorota.gryko@icho.edu.pl

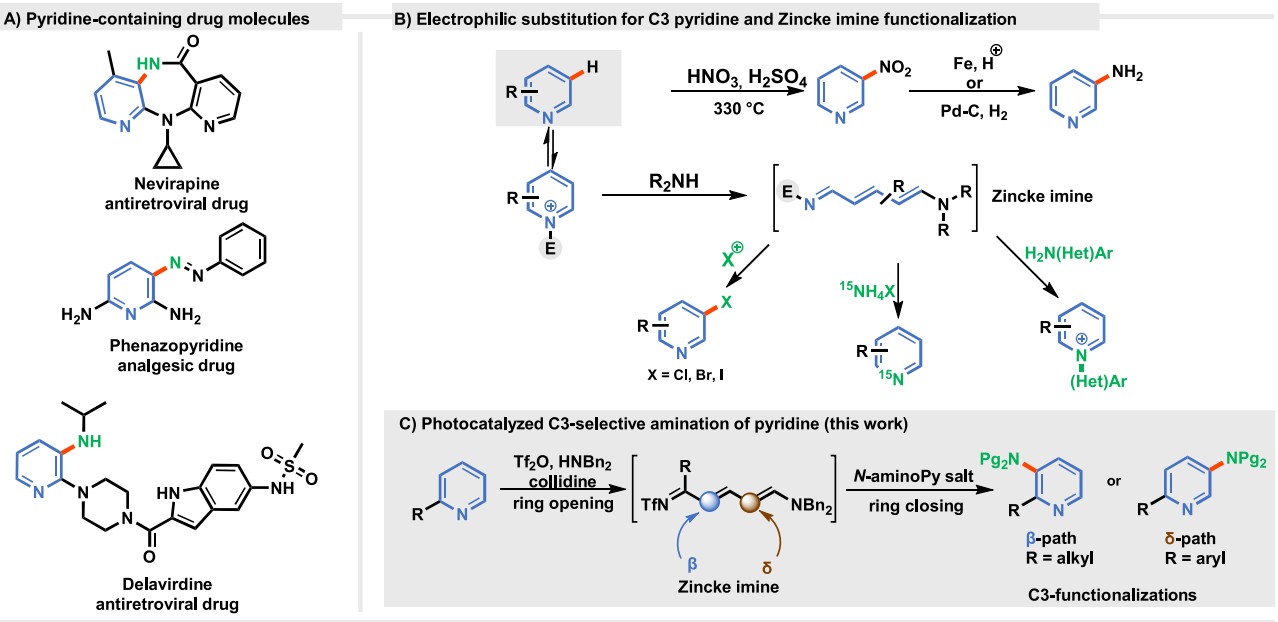

**Fig. 1 | State-of-the-art in the synthesis of C3-aminopyridines. A** Representative drugs bearing the pyridine moiety and (**B**) Strategies enabling C-H functionalizations of pyridines at the C3 position. **C** This work – Photocatalyzed C3-selective amination of pyridine.

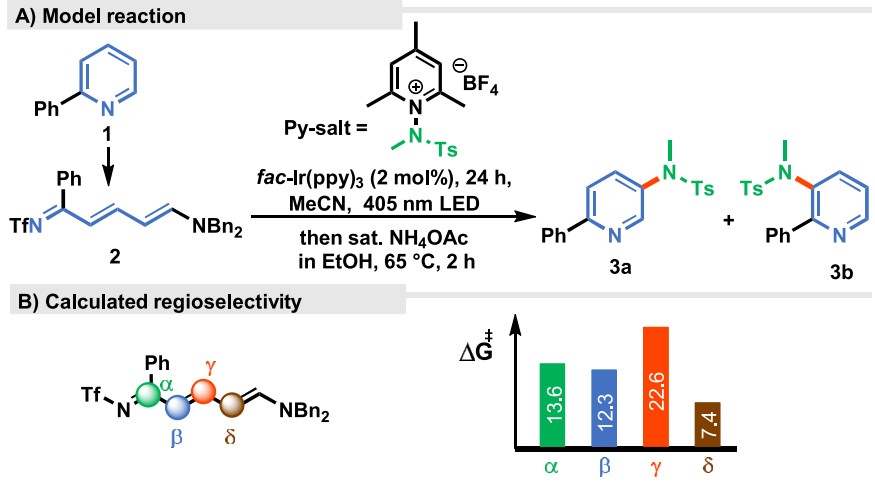

**Fig. 2 | Photochemical amination of pyridines. A** Model reaction of Zincke imine **2** with *N*-aminopyridinum salt. **B** Calculated Gibbs free energies for the regioselective reactions.

ethers, we hypothesized that selective C3-amination of pyridines could be achieved in a photocatalytic manner. Indeed, here we demonstrate that N-centred radicals, generated from *N*-aminopyridinum salts in a photochemical manner, react with Zincke imines to regioselectively produce, after rearomatization, C3-amino pyridines (Fig. 1C).

## Results and discussion
### Reaction development
Based on the reported data, in the initial phase of our study, model 2-phenyl Zincke imine **2**, obtained from 2-phenylpyridine (**1**)[29], was reacted with an aminopyridinium salt in the presence of *fac*-Ir(ppy)₃ under blue light irradiation to give, after the consecutive ring closure, desired product **3** (for details, see SI), but in a low yield (Fig. 2A).

DFT calculations revealed that the formation of C-N bonds at the δ position should be predominant over the β position due to the lower transition state energy (Fig. 2B, *vide infra* for the full free energy profile). Thorough optimization studies enabled a significant

improvement in yield (Table 1, entry 1). The optimal conditions are as follows: under an argon atmosphere, a mixture of 2-phenyl Zincke imine **2** (1 equiv.) and *N*-aminopyridinium salt (1.5 equiv.) in the presence of *fac*-Ir(ppy)₃ (2 mol%) in a MeCN/DMSO mixture (v/v 1:1) (0.006 M) was irradiated (LED, 405 nm) at 0 °C for 24 h. The subsequent treatment with saturated NH₄OAc in EtOH in a one-pot manner gave the desired rearomatized C3-aminopyridine **3** as a mixture of two regioisomers **3a** and **3b** in 99% yield (5:1, entry 1). Among other Zincke imines tested, only the *N*-benzylaniline derivative yielded the desired product, although a significant drop in both yield (26%) and regioselectivity to a 2:1 ratio was observed (for further information, see SI). The use of either sole DMSO or MeCN diminished the yield (entries 2 and 3). Previous studies have demonstrated that reduction of *N*-aminopyridinium salts to N-centred radicals by Ir(III) in the excited state involving single electron transfer (SET) can be performed under blue LED irradiation[38,48]. Intriguingly, in our case, the reaction yield decreased significantly to

## Table 1 | Conditions for selective C3-amination of pyridines via Zincke imine

| Entry | Variation form optimal conditions[d] | Yield (%) 3a and 3b (%)[a] | 3a:3b ratio[a] |
|---|---|---|---|
| 1 | None | >99; >95[b], 95[c] | 4.8:1 |
| 2 | MeCN | 66 | 2.6:1 |
| 3 | DMSO | 48 | 5.2:1 |
| 4 | Blue LED (455 nm) | 16 | n.d. |
| 5 | 1.1 equiv. of Py salt | 68 | 5.8:1 |
| 6 | 0.0125 M | 61 | 3.1:1 |
| 7 | No PC | 6 | n.d. |
| 8 | No light | Traces | n.d. |
| 9 | No light + no PC | 0 | n.d. |

[a]Yields determined by GC-FID analysis of crude reaction mixtures.
[b]Yield determined by $^1$H NMR analysis of the crude reaction mixture with $CH_2Br_2$ as internal standard.
[c]Isolated yield.
[d]Optimal conditions: 0.05 mmol scale (2, $c$ = 0.006 M), 0.075 mmol (1.5 equiv. Py salt), 2 mol% $fac$-Ir(ppy)$_3$, violet LED (405 nm), 24 h, 0 °C, DMSO/MeCN (v:v 1:1), then sat. NH$_4$OAc in EtOH, 65 °C, 2 h.

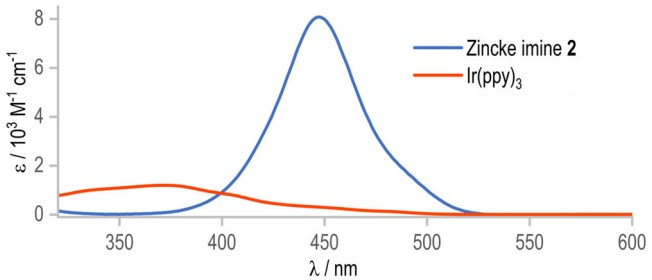

**Fig. 3 | UV-Vis spectrum of Zincke Imine 2 and $fac$-Ir(ppy)$_3$.** Spectra measured in (MeCN), blue line – Zincke imine **2**, orange line – $fac$-Ir(ppy)$_3$.

16%, probably due to strong absorption around 450 nm in the UV-Vis spectrum of Zincke imine **2** (entry 4, Fig. 3). As a result, violet light (405 nm) irradiation proved a crucial factor for the excitation of the Ir-catalyst and thus the generation of amidyl radicals in the presence of Zincke imines. A lower excess of the pyridinium salt (1.1 equiv.) did not ensure a full conversion of the starting material (entry 5). Other mono- and di-protected N-aminopyridinium salts were less effective under the developed conditions (for further information, see SI). Dilute conditions were deemed necessary, as a significant decrease in the yield was observed in the case of concentrated solutions (entry 6). Control experiments, without the photocatalyst or light clearly indicated that the reaction is a photochemically induced process (entries 7–9).

## Scope and limitations

The scope with respect to the pyridine was then examined (Fig. 4). Zincke Imines derived from 2-phenyl-substituted pyridines bearing electron-donating groups (EDG) (e.g., methoxy, methyl) on the phenyl ring are well tolerated giving products **4**–**12** in high yields, though the yield diminishes when substituents impose a steric hindrance around the newly forming bond, for example two methoxy groups at *ortho*-positions (**7a**). In these cases, harsher conditions are required for effective ring closure (using a microwave with higher temperature; for details, see SI). However, having only one substituent in the *ortho* position such as 2-methylthio-substituted pyridine, allowed for the synthesis the desired product **8a** in 88% yield and high regioselectivity. Other 2-aryl derivatives, such as biphenyls (**13**–**15**), phenanthrene (**17**), and pyrene (**18**) furnishes similar results,

but we observed a reduced yield for the naphthalene derivative (**16**) due to incomplete conversion of the reaction. On the other hand, electron-withdrawing groups (e. g., -CN, NO$_2$) are not only well tolerated, but also have a beneficial impact on the regioselectivity of the reaction. For example, the reaction with 2-(4-nitrophenyl)pyridine affords exclusively product **20a** in 98% yield. 2-Heteroaryl substituted pyridines provide aminated products **24**–**29** in satisfactory yields (35–86%). The low yield for the substrate bearing the N-Boc-pyrrole moiety results from partial deprotection of the carbamate group under the acidic condition. Furthermore, the influence of alkyl groups at position 2 was examined. For a series of 2-alkylpyridines (methyl, ethyl, hexyl, and benzyl), decreased yields were observed under standard conditions. We fine-tuned the reaction conditions and found that the exclusive use of MeCN, alongside with a higher power of the light source, effectively increased the efficiency of the C3-amination reaction with 2-alkylsubstituents (for details, see SI). Interestingly, the regioselectivity switches, in this case an amidyl radical attacks Zincke imine preferentially at the $\beta$ position therefore, the resultant pyridine bears substituents at positions C2 and C3 in contrast to 2- phenylpyridine, for which they occupy positions C3 and C5. An exception are isopropyl and methoxy-substituted benzyl derivatives that form aminated products **33** and **35** with almost no selectivity.

To further demonstrate the synthetic utility, the one-pot amination reaction involving three consecutive steps was attempted. The process is compatible with the model reaction, after three steps, the total yield equals 55% and the regioselectivity did not erode (Fig. 5).

## Mechanistic studies

To gain a better understanding of the reaction developed, we conducted a series of control experiments. They indeed confirmed the relevance of both the light and the catalyst. The radical nature of the mechanism was confirmed by an experiment with the addition of TEMPO, which completely halted the reaction (Fig. 6A). DMPO, a radical spin trap was added to the reaction mixture, which formed an adduct with the radical generated from the N-aminopyridinium salt as confirmed by ESI-MS analysis (Fig. 6B). Furthermore, kinetic experiments revealed that amidated Zincke imine **2a** forms gradually over 12 h (Fig. 6C). Due to its instability, the isolated yield was only 15% but allowed us to confirm its structure by NMR spectroscopy and X-ray crystallography (Fig. 6D).

Based on data from our previous work and literature[49,50], along with our control experiments and DFT and DLPNO-CCSD(T) calculations (Fig. 7A, for more computational details, see SI), we propose a light-induced radical formation of N-aminopyridinium salt ($E_{1/2}$ = −0.70 V vs Ag/AgCl) via Single Electron Transfer (SET) from the excited [Ir(III)] photoredox catalyst (Fig. 7B). N-aminopyridinium salt is reduced and generates radical **A** (−24.4 kcal/mol). This radical undergoes fragmentation to form an N-centred radical **B** (−52.6 kcal/mol) and collidine as a byproduct through a low energy transition state (**TS-1**, 4.1 kcal/mol). The resulting electrophilic N-centred radical **B** reacts with the Zincke imine derivative at the $\delta$-position ($\Delta G^\ddagger$ = 7.4 kcal/mol), leading to the formation of a C-N bond and the corresponding intermediate **C** (−52.0 kcal/mol).

Then, intermediate **C** can be easily oxidized by the Ir-(IV) catalyst to form cation derivative **D** ($\Delta G$ = −64.1 kJ/mol) with the regeneration of the ground state of the Ir-(III) catalyst. Finally, deprotonation of cation **D** with collidine affords product **E** ($\Delta G°$ = −106.7 kcal/mol). From this point, ring cyclization from product E to final product **3a** occurs via a deprotection/ring closure sequence with a barrier of 15.1 kcal/mol (see Supplementary Information and Supplementary Data 1 for details).

The reaction mechanism was also evaluated for the attack of the N-centred radical at position $\alpha$, $\beta$ and $\gamma$ (Fig. 2B, see SI for the selectivity

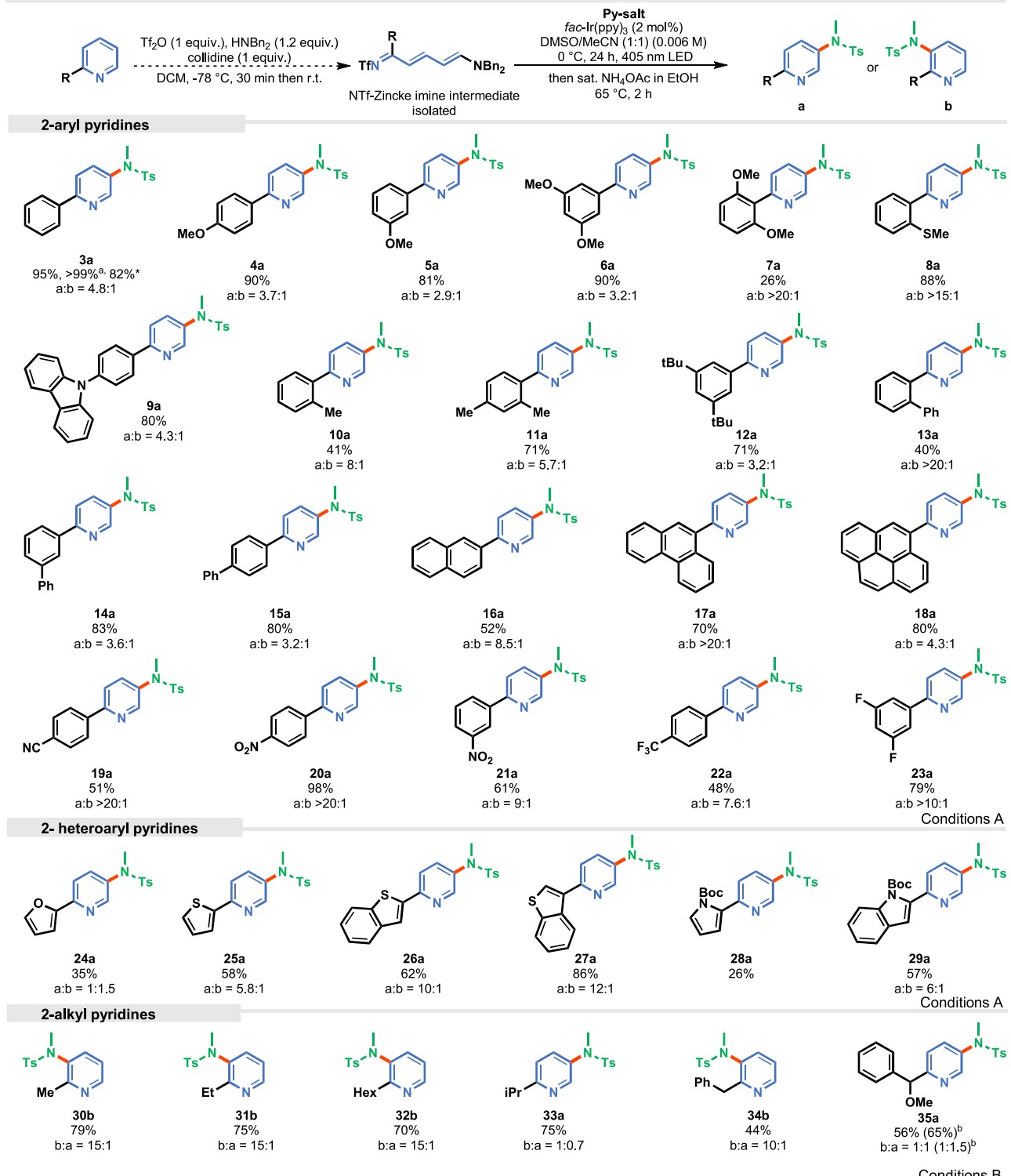

**Fig. 4 | Scope of C3-amination of pyridines.** The yields given are isolated unless otherwise stated. [a] Yield determined by GC-FID analysis of crude reaction mixtures; [b] Reaction was carried out under Conditions A; The ratio was determined by the GC-FID method or by ¹H NMR analysis of crude reaction mixtures; for each case a major isomer is drawn. Conditions A: 1) Zincke imine (0.05 mmol), *fac*-Ir(ppy)₃ (2 mol%), *N*-aminopyridinium salt (1.5 equiv.), DMSO/MeCN (1:1), 0 °C, 24 h, 405 nm LED (2.4 W); 2) sat. NH₄OAc in EtOH, 65 °C, 2 h. Conditions B: 1) Zincke imine (0.05 mmol), *fac*-Ir(ppy)₃ (2 mol%), *N*-aminopyridinium salt (1.5 equiv.), dry MeCN, 0 °C, 24 h, 405 nm LED (4.8 W); 2) sat. NH₄OAc in EtOH, 65 °C, 2 h. *Zincke imine (0.2 mmol scale).

evaluation). At $\beta$, the reaction can be competitive, as demonstrated by the experimental observation of the minor isomer. However, at $\gamma$, the radical attack is much higher in energy (22.6 kcal/mol) and at $\alpha$ the activation free energy is similar to $\beta$ (13.6 kcal/mol), but the resulting

radical cannot be oxidized, preventing the turnover of the catalytic cycle and resulting as a nonproductive reaction pathway.

In the final stage, further synthetic transformations of synthesized pyridines were then explored: the deprotection of the tosyl group and

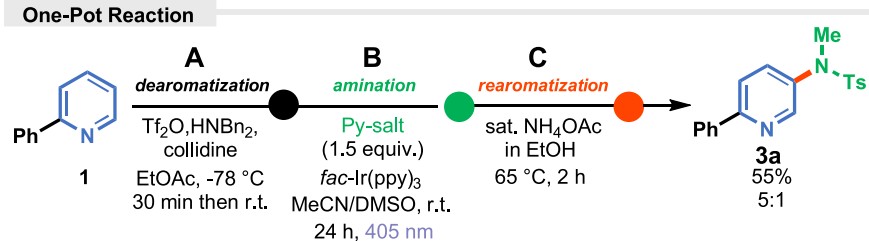

**Fig. 5 | One-pot *meta*-amination of 2-phenylpyridine (1). A** Dearomatization: Tf₂O, HNBn₂, EtOAc, −78 °C to r.t.); (**B**) Amination: Py-salt, *fac*-Ir(ppy)₃, MeCN/DMSO, r.t., 24 h, 405 nm LEDs; (**C**) Rearomatization; sat. NH₄OAc₍EtOH₎, 65 °C, 2 h).

**Fig. 6 | Mechanistic studies. A** Radical trap experiment with TEMPO. **B** Radical trap experiment with DMPO. **C** Kinetic reaction profile of the model reaction. **D** X-Ray structure of aminated Zincke imine **2a**.

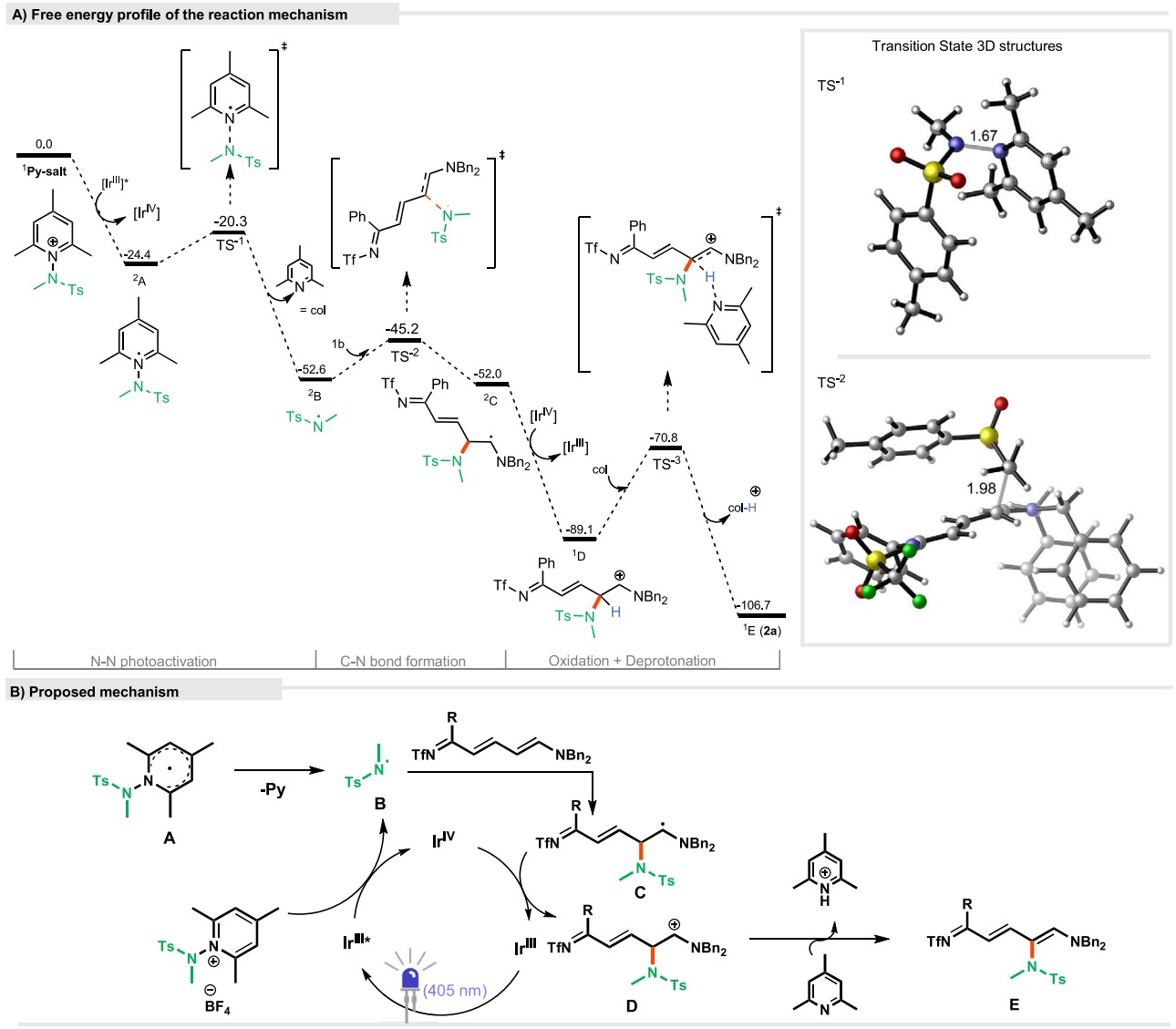

**Fig. 7 | Theoretical studies. A** Free energy profile of the reaction mechanism for the major product at SMD(MeCN) DLPNO-CCSD(T)/Def2TZVPP//ωB97xD/Def2SVP level of theory. Free energies in kcal/mol and bond distances in Å. **B** Proposed mechanism.

peripheral editing of the pyridine ring. Both regioisomers underwent tosyl group cleavage with high yields, providing free NH groups (**36** and **37**) (Fig. 8A).

The resulting pyridine derivative was then readily acylated with acyl halide, forming amide **38** in 80% yield (Fig. 8B). Furthermore, the synthesis of *N*-oxide (**39**) and *N*-methylpyridinium (**40**) derivatives was successfully achieved using *m*CPBA and MeI, respectively, in good yields (Fig. 8C, D). The sequential C3- and C5-difunctionalization of pyridine was also investigated using 2-phenyl Zincke imine **2**. The resulting aminated intermediate underwent a subsequent functionalization through an bromination with *N*-bromosuccinimide,[29] yielding regioselectively brominated pyridine **41a** in 65% yield after the rearomatization (Fig. 8E).

In summary, we have developed a photochemical methodology for the peripheral editing of the pyridine core that relies on a dearomatization/aromatization strategy. After activation of a pyridine as a Zincke imine intermediate, it reacts with an electrophilic N-centred radical generated from *N*-aminopyridinium salts in a photochemical manner. The amino-derivative formed, after aromatization, furnishes the desired product. Importantly, the C–N bond formation occurs predominantly at C3 position. Depending on the nature of the C2-

substituent (aryl versus alkyl) in the starting material, C3-, C5- or C2-, C3-functionalized pyridines are formed. The method is characterized by mild reaction conditions, scalability, pyridine as the limiting reagent, and excellent regioselectivity. Importantly, it can be performed in a one-pot fashion. DFT calculations confirms the preferential attack at the δ position, and the reaction mechanism consists of the following reaction steps: generation of N-centred radical via Ir-mediated reduction of the *N*-aminopyridinium salt, selective addition of N-radical and oxidation/deprotonation of the resulting radical species.

Our work opens new photochemical avenues in the peripheral editing of the pyridine scaffold via Zincke imine. Further work is currently undergoing in our laboratory.

## Methods
### Synthetic procedures and compound characterization
Photochemical reactions were carried out in the UOSlab Miniphoto photoreactor. Detailed synthetic procedures, including reaction conditions, yields, NMR spectra, high-resolution mass spectrometry, and X-ray crystallographic data, are given in the Supplementary Information.

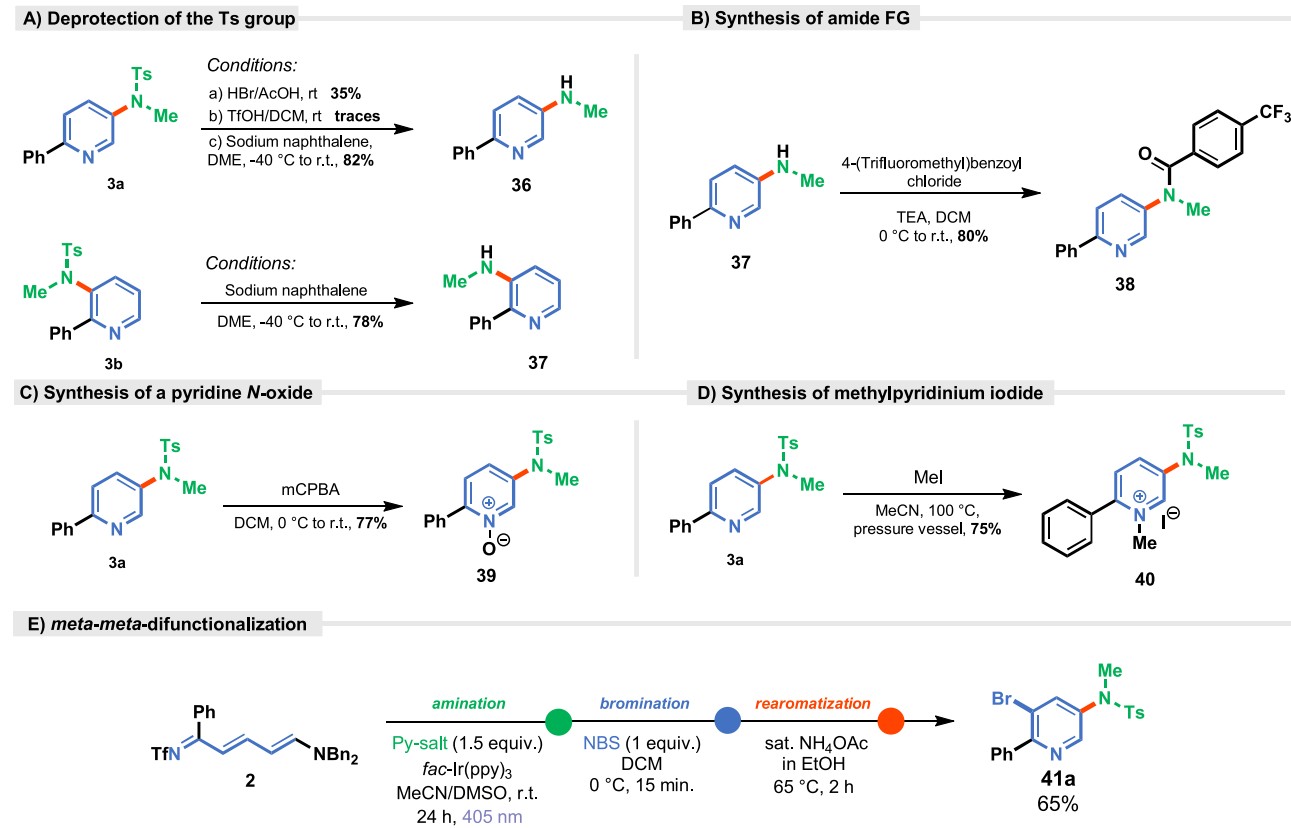

**Fig. 8 | Synthetic applications. A** Deprotection of the Ts group under different reaction conditions, **B** Synthesis of amide functional group, **C** Synthesis of a pyridine *N*-oxide, **D** Synthesis of methylpyridinium iodide, **E** Formation of *meta-meta*-difunctionalized 2-phenylpyridine via Zincke Imine.

## General protocol for photoamination and closure of the 2-aryl Zincke imines

Zincke imine (0.05 mmol), Py-salt (0.075 mmol), *fac*-Ir(ppy)₃ (0.67 mg, ~10 μmol, 2 mol%) were placed in the closed-cup vial and MeCN (4 ml) and DMSO (4 ml) were added through the septum. The reaction mixture was placed in ultrasound bath and degassed by bubbling argon through the solution for 15 min. The vial was then moved to the photoreactor and irradiated with violet light (2.4 W) for 24 h maintaining temperature between 0 °C to 5 °C with a dedicated cooling system. After the indicated time, a saturated NH₄OAc solution was added in anhydrous ethanol (2 ml) and reaction was heated up to 65 °C for 2 h. DMSO and an excess of NH₄OAc were removed by extraction (AcOEt/ H₂O). The organic phase was dried over anhydrous sodium sulphate and evaporated with silica gel (dry load for the preparation of the sample for flash chromatography). The pure products were isolated by flash chromatography in the hexanes/AcOEt gradient.

## General procedure for the functionalization of 2-alkyl Zincke imines

Zincke imine (0.05 mmol), Py-salt (0.075 mmol), *fac*-Ir(ppy)₃ (0.67 mg, ~10 μmol, 2 mol%) were placed in the closed-cup vial and MeCN (8 ml) were added through the septum. The reaction mixture was placed in ultrasound bath and degassed by bubbling argon through the solution for 15 min. The vial was then moved to the photoreactor and irradiated with violet light (4.8 W) for 24 h, maintaining a temperature between 0 °C and 5 °C. After the indicated time, the saturated NH₄OAc solution in anhydrous ethanol was added (2 ml) and the reaction mixture was heated up to 65 °C for 2 h. The excess of NH₄OAc was removed by extraction (AcOEt/ H₂O). The organic phase was dried over anhydrous sodium sulphate and evaporated with silica gel (dry load for the preparation of the sample for

flash chromatography). The pure products were isolated by flash chromatography in hexanes/AcOEt gradient.

## Computational details

DFT calculations were carried out using the G16 programme package using the ωB97xD functional. Geometry optimisations and frequency calculations were computed with the Def2SVP basis set without symmetry restrictions. The nature of all the stationary points was characterized by frequency calculations as minima (no imaginary frequencies) or transition states (one imaginary frequency). Transition states were relaxed to reactants and products, and IRC calculations were performed to further validate the connectivity. Additionally, the solvation energy was obtained from single-point calculations using ωB97xD/Def2TZVPP and the implicit solvent model (acetonitrile). The solvation free energy was then obtained by the difference between the energy calculated with the SMD model – the energy in gas phase. The standard state was corrected from 1 atm to 1 M by adding 1.89 kcal/mol when needed.

The potential energies were further refined using the DLPNO-CCSD(T) method in ORCA. Combination of Def2TZVPP and Ri-C auxiliary basis set (Def2-TZVPP/C) and RIJCOSX (Def2/J). The tightSCF option was also selected.

## Data availability

The authors declare that the data supporting the findings of this study are available within the article and its Supplementary Information (experimental details, NMR spectroscopic data, X-ray crystal information), Supplementary Data 1 (Cartesian coordinates of DFT calculated geometries) or from the corresponding authors upon request. Crystallographic data for the aminated Zincke imine **2a** reported in this article has been deposited at the Cambridge Crystallographic Data Centre under deposition numbers CCDC 2369412. Copies of the data

can be obtained free of charge via https://www.ccdc.cam.ac.uk/structures/.

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

## Acknowledgements

This work is supported by the National Science Centre, Poland (MAESTRO UMO-2020/38/A/ST4/00185, K.F. Sz., P.B., D.G.). I.F.-A thank the project PID2021-126075NB-I00 funded by MCIN/AEI/10.13039/501100011033 and the European Union "Next Generation EU"/PRTR. He also acknowledged MCIN/AEI/ 10.13039/501100011033 for the "Ramón y Cajal" scholarship (RYC2022-035776-I).

## Author contributions

D.G. conceived the project and designed the initial experiments. K.F. Sz., P.B., A.P. developed methodology and performed experiments regarding the synthesis and characterization of all the compounds. I.F.A. designed the computational study and. D.L. performed the calculations. All authors analysed the data, discussed the results, and commented on the manuscript. K.F.Sz. and P.B. contributed equally to this work.

## Competing interests

The authors declare no competing interests.
