## [Transparent Peer Review file · Nature Communications]

Photochemical C3-Amination of Pyridines via Zincke Imine Intermediates

Corresponding Author: Professor Dorota Gryko

Version 0:

Reviewer comments:

Reviewer #1

(Remarks to the Author)

Comments:

This manuscript deals with a photochemical C-3 amination of pyridines via Zincke imine intermediates. The manuscript is written well and the results were supported by mechanistic and extensive theoretical studies. Moreover, the scope of pyridine derivatives were well demonstrated and overall this work is of general interest to researchers from all areas of organic synthesis.

The manuscript could be considered for Nature Communication after addressing the below concern

1. Author should test the scope of nitrogen centered radical with other functional groups such as benzoyl, carbamate, phthalimide etc.
2. It is recommended to perform subsequent derivatization of product via deprotection of tosyl group.
3. How did the authors come up with dibenzyl amine for zincke imine intermediate synthesis? Does any other secondary amines were tested for this reaction? If not then does dibenzyl group has any role in observed regioselectivity.

Reviewer #2

(Remarks to the Author)

The author developed a photochemical methodology for the peripheral editing of the pyridine core that relies on a dearomatization/aromatization strategy. This strategy further enriches the pyridine C3 (or C5) amination reaction and is of significant importance. I am supportive of publishing this study in Nature Communications subject to the following revisions:

1. The author mentioned that "electron-withdrawing groups (e. g., -CF₃, difluoro) are not only well tolerated, but also have a beneficial impact on the regioselectivity of the reaction." Personally, I find that illustrating with examples cyano (CN-) and nitro (NO₂-) groups are more advantageous than CF₃ and difluoro would be more enlightening
2. While exploring the applicability of the reaction, the author investigated whether aminopyridinium salts, such as common primary amines, dialkylamines, aromatic amines, or N-methylamide compounds pyridinium salts could be used.
3. To facilitate readers' understanding with clarity and simplicity, the TEMPO trapping experiment mentioned on the line 130 by the author can be illustrated in a diagram.
4. The sequence of the NMR spectrum for compounds 16a, 17a, and 18a was incorrect, and there was an issue with the integration of compound 35b.

Reviewer #3

(Remarks to the Author)

Crystallographic data looks good. I'll let the other reviewers comment on the science since that is not my task.

Reviewer #4

(Remarks to the Author)

This paper reports a very interesting photochemical C3-amination of pyridines. I am especially interested in the experimental outcome in Figure 4.

The authors suggest that the low yield of 7a might be caused by the steric effect. But the other sterically-hindered reagents do not always give low yield. I think the authors should pay more effort to propose a more convincing reason. On the other hand, the 2-alkylpyridines show very different regioselectivity, with 30b-32b gives C3-selectivity and 33a gives C5-selectivity. The authors should added reasonable analysis and maybe also DFT calculations to rationalize the observed selectivity. Why iPr is different from the other alkyl and benzyl cases? Finally, in addition to the DFT calculations on the conversion from 1 to 2, the authors should also add calculations on the energy changes of the conversion from 2 to 3, and also to explain why the aryl and alkyl substituent results in different selectivity.

Version 1:

Reviewer comments:

Reviewer #1

(Remarks to the Author)

The questions from this reviewer are fully addressed by these authors. I think that the manuscript in its present form is appropriate for publishing in Nature Communications.

Reviewer #2

(Remarks to the Author)

The authors report a mild regioselective C3 amination method relying on the photochemical reaction of zinc imine with amide radicals generated from N-aminopyridinium salts. They elucidate the C-3 regioselectivity of the reaction. This work not only fills a gap in previous studies but also expands the chemical space for selective skeletal and peripheral editing of pyridines. The experimental data are comprehensive and reliable, and the research methods are scientifically sound. The author effectively addressed and resolved the questions raised by the reviewers, and it is recommended for publication in Nature Communications.

Reviewer #4

(Remarks to the Author)

The authors have responded to my comments. But I don't think the calculation in Page 68 of SI reasonably explained the selectivity issue. This issue could be addressed by a more detailed study. Therefore I would suggest to remove the related discussion and results to avoid misleading to readers.

Manuscript ID: Nature Communications manuscript **NCOMMS-24-63500**

REVIEWER 1:

Comments:

- 1. Author should test the scope of nitrogen centered radical with other functional groups such as benzoyl, carbamate, phthalimide etc.*
- 2. It is recommended to perform subsequent derivatization of the product through deprotection of the tosyl group.*

We have tested three different conditions for the deprotection of the tosyl group and finally we found the conditions where we were able to remove tosyl group with 82% yield from **3a** isomer and with 78% yield for **3b** isomer (additional scheme Fig.8 is added to the manuscript, page 10). We have provided a proper analysis of both products ¹H, ¹³C, HRMS in the supporting information, as well as the protocol for the deprotection. Additionally, we have performed several derivatizations of amidated product **3a** (or detosylated product **36**, Fig. 8)

 - 1) We performed amidation of deprotected (amine) moiety for compound **36**
 - 2) We have performed the transformation of product **3a** into the N-oxide
 - 3) We have performed alkylation (with MeI) nitrogen in the pyridine moiety for the product **3a**
 - 4) We performed difunctionalization of 2Ph-Zincke Imine **2** (amination-bromination), obtaining valuable pyridine derivatives with good total yield (>65%)
- 3. How did the authors come up with dibenzyl amine for zincke imine intermediate synthesis? Does any other secondary amines were tested for this reaction? If not then does dibenzyl group has any role in observed regioselectivity.*

The preparation and optimisation of the Zincke imine' structure were published by McNally's group (*Science*, **2022**, 378, 773). The authors optimized the synthesis and final closing of Zincke Imines using different amines, concluding that dibenzylamine is the most suitable for this process. *However, we had prepared two different 2Ph Zincke imines starting from 2-phenylpyridine to check if the attached amine moiety influences amidation and further rearomatization. Our conclusions are presented in the revised version of the manuscript (page 4, line 6).*

REVIEWER 2:

Comments:

- 1. The author mentioned that "electron-withdrawing groups (e. g., -CF₃, difluoro) are not only well tolerated, but also have a beneficial impact on the regioselectivity of the reaction." Personally, I find that illustrating with examples cyano (CN-) and nitro (NO₂-) groups are more advantageous than CF₃ and difluoro would be more enlightening.*

The authors thank the reviewer for this meaningful suggestion. The appropriate changes have been made.

2. While exploring the applicability of the reaction, the author investigated whether aminopyridinium salts, such as common primary amines, dialkylamines, aromatic amines, or *N*-methylamide compounds pyridinium salts could be used.

We have tested *N*-aminopyridinium salts and only -NMeTs-pyridinium salt gave reasonable conversion of the starting material and it was used for further optimization studies. The same results were later obtained under the developed conditions (all information was added as a table in SI (page 64) and mentioned in the manuscript (page 4, line 15). We do, however, believe that if necessary detailed tuning of the reaction conditions would allow for the synthesis of functionalized pyridines from other salts. This is corroborated by our preliminary studies, we performed short optimization studies for NHTs salt and found traces of the desired product.

3. To facilitate readers' understanding with clarity and simplicity, the TEMPO trapping experiment mentioned on the line 130 by the author can be illustrated in a diagram.

The TEMPO trapping experiment was added to the manuscript (page 8, Fig. 6A).

4. The sequence of the NMR spectrum for compounds 16a, 17a, and 18a was incorrect, and there was an issue with the integration of compound 35b.

The authors thank the reviewer for his comment. We corrected our mistake and we changed the order of the spectra. Thanks to his remark, we have realized that compound **35b** is pure and exists as a mixture of rotamers. We have added ¹H and ¹³C spectra in both CDCl₃ and CD₃OD, and measured ¹H and ¹³C spectra in DMSO-d₆ at elevated temperature (all the data are available in SI, page 207-212).

REVIEWER 3:

We thank the referee for his statement.

REVIEWER 4:

Comments:

1. I am especially interested in the experimental outcome in Figure 4. The authors suggest that the low yield of **7a** might be caused by the steric effect. But the other sterically-hindered reagents do not always give low yield. I think the authors should pay more effort to propose a more convincing reason.

We have found that even for non-functionalized compound **7**, the conversion for the ring close under the standard conditions is not full. Compound **7a** is the only derivative bearing two *ortho*-substituents in the phenyl ring attached to the pyridine moiety, which causes problems in the ring-close step. We have not found significant decomposition of functionalised Zincke derived from compound **7-int**, and the full conversion of compound **7** was also observed; thus, we postulate that the ring-closing step is the limiting step. Furthermore, we did not observe the same effects for 2-thiophene or 2-indole based derivatives.

2. On the other hand, the 2-alkylpyridines show very different regioselectivity, with **30b-32b** gives C3-selectivity and **33a** gives C5-selectivity. The authors should add reasonable analysis and maybe also DFT calculations to rationalize the observed selectivity. Why *iPr* is different from the other alkyl and benzyl cases?

Regarding the selectivity of Ar vs. alkyl substituents, the relative free energies for C3 vs. C5 selectivity for the Ph, Me and *iPr* substituted substrates were added to the supporting information (page 68, Si). The small alkyl substituent, like Me, does not have a steric effect in the radical approximation, and we did not find a barrier in the potential energy surface (as shown in the relaxed scan of the C-N bond). We estimate the entropic

barrier calculating a long range adduct but a more in depth studies, including BOMD would be required to precisely quantify the ratio, which we consider to be out of the scope of the manuscript. On the contrary, the iPr group hinders the approximation of the radical to the C3 position and the relative barrier increased. However, the relative barrier of C3 vs C5 is clearly closer than for the Ar group, which is in agreement with the experimental observation of lower selectivity.

- 3. Finally, in addition to the DFT calculations on the conversion from 1 to 2, the authors should also add calculations on the energy changes of the conversion from 2 to 3, and also explain why the aryl and alkyl substituent results in different selectivity.*

We have also calculated the conversion of **2** to **3**. Although the ring-closing mechanism has been widely proposed in the literature, we did not find any computational study of the mechanism. In fact, attempting the proposed mechanisms (direct ring closing or protonation/ring closing sequence) resulted in a very high free energy barrier. Interestingly, under saturated NH₄OAc, the deprotection of the Tf group is highly exergonic and generates an active intermediate that can cyclize through a low transition state (15.1 kcal/mol) (page 67, SI. From this point on, proton migration and HNBn₂, recover the aromaticity, and generate product **3** exergonically.

Manuscript ID: Nature Communications manuscript NCOMMS-24-63500A

We thank the referees for their kind and meaningful remarks regarding our submission, which helped us improve our work, and for suggesting its publication in *Nature Communications*. We have done our best to address all concerns according to the suggestions of the reviewers. We have also resolved all technical issues as requested. A detailed description of all changes made is listed below.

REVIEWER 1:

Comments:

1. The questions from this reviewer are fully addressed by these authors. I think that the manuscript in its present form is appropriate for publishing in *Nature Communications*.

We thank the reviewer for his recommendation.

> **REVIEWER 2:**

Comments:

1. *The author effectively addressed and resolved the questions raised by the reviewers, and it is recommended for publication in Nature Communications.*

We thank the reviewer for his recommendation.

REVIEWER 4:

Comments:

1. *The authors have responded to my comments. But I don't think the calculation in Page 68 of SI reasonably explained the selectivity issue. This issue could be addressed by a more detailed study. Therefore I would suggest to remove the related discussion and results to avoid misleading to readers.*

We have done our best with the theoretical studies but if this is not sufficient, we have decided to follow the reviewer's suggestion and removed this part.

We hope with these new amendments our work will be as acceptable for publication in *Nature Communications*.

Yours sincerely,

Prof. Dorota Gryko